# A Simple Telemetry Sensor System for Monitoring Body Temperature in Rabbits—A Brief Report

**DOI:** 10.3390/ani13101677

**Published:** 2023-05-18

**Authors:** Yajie Chen, Manabu Niimi, Lan Zhang, Xiangming Tang, Jian Lu, Jianglin Fan

**Affiliations:** 1Department of Molecular Pathology, Faculty of Medicine, Graduate School of Medical Sciences, University of Yamanashi, Chuo 409-3898, Japan; yajie_wuyi@yeah.net (Y.C.);; 2National Institute of Advanced Industrial Science and Technology, Tsukuba 305-8564, Japan; 3Guangdong Province Key Laboratory, Southern China Institute of Large Animal Models for Biomedicine, School of Biotechnology and Health Sciences, Wuyi University, Jiangmen 529020, China

**Keywords:** implantable sensor, body temperature, rabbits, telemetry

## Abstract

**Simple Summary:**

Measurement of body temperature is essential for the study of metabolic changes and inflammation using experimental animals. Although measuring the rectal temperature is often used, this method is not only time-consuming but also stressful for animals. In addition, it is not possible to use this method to make a constant measurement of body temperature. In the current study, we developed an all-in-one packaged telemetry system, which consists of a temperature sensor, transmitter, balun, and a 3-axis accelerometer. The telemetry sensor was easily implanted subcutaneously in rabbits housed in the animal facility while temperature changes were continuously recorded through a personal computer. Our telemetry sensor system may be useful for monitoring body temperature in experimental animals such as rabbits.

**Abstract:**

Continuous body temperature measurement is an important means of studying inflammation and metabolic changes using experimental animals. Although expensive telemetry equipment for collecting multiple parameters is available for small animals, readily used devices for mediate- or large-sized animals are rather limited. In this study, we developed a new telemetry sensor system that can continuously monitor rabbit body temperature. The telemetry sensor was easily implanted subcutaneously in rabbits housed in the animal facility while temperature changes were continuously recorded by a personal computer. Temperature data obtained by the telemetry was consistent with the rectal temperature measured by a digital device. Analysis of body temperature changes of unstrained rabbits, either under the normal condition or fever induced by endotoxin confirms the reliability and usefulness of this system.

## 1. Introduction

Measurement of body temperature along with heart rate, blood pressure, and physical activity is essential for the study of metabolic changes and inflammation using humans and animals because body temperature is one of the fundamental parameters for assessing the health status [1,2]. The body temperature can not only evaluate the healthy or inflammatory status but also can predict the further death [3]. Compared with the body surface temperature, the core temperature, referred to the temperature of the hypothalamus or deep body sites, was considered more precise and stable [4]. The pulmonary artery catheterization, a highly invasive procedure, is considered a gold standard to measure the core body temperature [5]. The rectal temperature measured by a thermometer is less invasive and the most widely used to monitor body temperature in young children (<3 y) and experimental animals [6,7]. However, this method is limited by stress caused to animals and it is not possible to monitor body temperature constantly for a long time. Infrared thermometry, a total noninvasive and noncontact technology, has been used to monitor the surface temperature of humans, rhesus macaques, and pigs [6,8], but such a method is considered less accurate [3]. Wireless body temperature measurements using implanted digital probes were more accurate [9]. The digital probes implanted subcutaneously are commonly used for a long time and consistently monitor the core body temperature [10]. Depending on the methods, the digital probes can be classified as transponders, telemetry transmitters, and data loggers [10]. Telemetry systems designed for small experimental animals (such as rodents) are well-established and commercially available, but such a system is usually expensive, complicated, and laborious [11]. Rabbits are a useful animal model for investigating metabolic syndrome and testing pyrogens although devices for measuring body temperature during the whole experiment period are rather limited [12,13,14,15]. Our coworkers developed a series of wireless sensors for use in social infrastructure, health care, and livestock [16,17,18]. They have produced an ultra-small and ultra-light telemetry transmitter for real-time monitoring of the body temperature consistently in mice and cows [16]. In the current study, we aimed to develop a new sensor system for measuring temperature in large animals such as rabbits. The implantable sensor was designed as an all-in-one packaged node using a 3D printer and consisted of a temperature sensor, transmitter, balun, and 3-axis accelerometer [16]. After implantation, the sensor could be tele-controlled by an ON/OFF switch to extend the lifespan of the power supply. To validate the usefulness of the telemetry sensor system, we subcutaneously implanted the sensor in rabbits. We examined the real-time body temperature changes either under normal conditions or inflammatory stimulation. Our results showed that this telemetry sensor system is reliable and simple, thus it may become a new tool for measuring body temperature in rabbits.

## 2. Materials and Methods

### 2.1. Telemetry Sensor System

The telemetry sensor system was made as described previously with a modification [16]. The current sensor node was characterized by miniatured size including both system block integration and physical interconnection, along with the lowest power consumption to extend the lifetime of the battery [16]. The system was essentially composed of four parts: an implantable sensor (TMP007, Texas Instruments Inc., Dallas, TX, USA), an antenna (A1101R09C, Anaren, Inc., East Syracuse, NY, USA), a receiver (SDBC-DK3, Silicon Labs, San Jose, CA, USA), and a personal computer (PC) (Figure 1A). A telemetry sensor structure, components, and specifications were depicted in detail in Figure 1B. The sensor contains a 3-axis accelerometer (ADXL362, Analog Devices Inc, Wilmington, MA, USA), which allows the measurement of physical activity and body position in addition to temperature. After packaging, the sensor node was validated by putting it into a temperature chamber to confirm its temperature accuracy and a shaker to check its response to acceleration. The surface of the sensor was coated with silicon. An antenna was placed on the rack of animal cages and linked with a personal computer (PC) via a receiver, which receives signals from the sensor (Figure 1A–D).

Body temperature changes were monitored by the sensor subcutaneously implanted in the posterior neck of rabbits (Figure 2). The temperature signal was automatically sent out through a radio transmitter and simultaneously captured by the receiver and transferred to a PC via a universal serial bus port. The open-source terminal emulator software (Tera Term, version 4.94 T. Terakihi, Tokyo, Japan) was used for data processing. The monitoring interval could be set even after implanting.

### 2.2. Implantation of the Sensor

Japanese White rabbits (Kbs:JW) were purchased from Kitayama Labes (Ina, Japan). We used female rabbits aged 22–34 months for the current experiments. The rabbits were individually housed under a light cycle with 12 h light/12 h dark at 23 °C room temperature. They were fed a standard regular diet (CR-3M; CLEA Japan, Tokyo, Japan) and water *ad libitum*. All animal experiments were approved by the Animal Care Committee of the University of Yamanashi (No. A3-50). For implantation of the sensors, three rabbits were anesthetized by intravenously injecting ketamine (Ketalar; Daiichi Sankyo, Tokyo, Japan) at 2.5 mg/kg and medetomidine (Domitor; Zenoaq, Koriyama, Japan) at 0.05 mg/kg. The fur of the posterior cervical region was shaved (Figure 2A). After disinfection with povidone-iodine, a skin incision (~3 cm) was made. A telemetry sensor disinfected by 70% ethanol solution was implanted subcutaneously (Figure 2B) and the incised skin was sutured (Figure 2C). The body temperature was measured in conscious rabbits for three consecutive days and recorded at 1 min intervals. To examine the response changes of body temperature to exogenous pyrogens mimicking acute inflammatory reaction, another three rabbits were injected with lipopolysaccharides (LPS) from *E. Coli* 055: B5 (Sigma-Aldrich, St. Louis, MO, USA) (2 μg/kg body weight) dissolved in saline solution via an ear vein and measured body temperature changes as described above. To compare the rectal temperature with the temperature measured by the sensor on the neck, the rectal temperature of the same conscious rabbits without immune challenge was measured every 6 h a day using a digital thermometer (DIGIFLASH; Génia, St. Hilaire de Chaléons, France). The mean values of triplicate measurement were used for analysis.

### 2.3. Statistical Analysis

Data are expressed as the mean ± SD. The Pearson correlation coefficient was performed using SPSS software version 21 (IBM Japan, Tokyo, Japan) for evaluating the correlation between rectal temperature using a digital thermometer and neck temperature measured by the sensor. *p* < 0.05 was considered statistically significant.

## 3. Results and Discussion

In the current study, we developed a new telemetry sensor system to measure the subcutaneous temperature in rabbits. The sensor system is characterized by having a radio frequency identification chip with a hybrid interface and neglectable power consumption. It was designed to enable researchers to switch ON/OFF and adjust measurement mode even after implanting the sensor in the animals. As shown in Figure 3, we measured three conscious rabbits for three consecutive days at 1 min intervals. We found that the data are consistent and repeatable in these rabbits, suggesting that a telemetry sensor system is a useful tool for monitoring body temperature in unstrained rabbits. The mean subcutaneous temperature was 38.44 ± 0.31 °C and the range of body temperature changes within 3 days was 1.44 ± 0.30 °C. The results are also consistent with others using different devices [19]. To validate the data collected by this sensor system, the rectal temperature of the same rabbits was measured by a digital thermometer, which is commonly used in rabbit studies [20]. The results showed that the subcutaneous body temperature was slightly lower than the rectal temperature (average −1.00 °C, ranging from −0.28 °C to −2.04 °C in 18 independent measurements). Moreover, temperatures measured by the two methods were well correlated (r = 0.925, *p* < 0.001) (Figure 4). Regardless of this, temperature accuracy may also be affected by different body sites chosen for the measurement. The battery loaded in the sensor in the current study can be sustained for more than 4 weeks (1 min interval as shown in the current study) or several months (if 10 min interval) dependent upon the recording time for different experiments. Therefore, it is possible to use this telemetry sensor system (even without replacing the battery) to perform longer-time experiments, such as metabolic syndrome, obesity, and diabetes [2,3]. Another advantage of this sensor system is that up to 20 rabbits can be simultaneously monitored using one telemetry sensor system. The current sensors could be repeatedly utilized after sterilizing and replacing a new battery.

We further investigated the usefulness of the sensor system for testing pyrogen-induced high fever. As shown in Figure 5, two rabbits showed rapid elevation of body temperature after LPS injection, consistent with the previous report [21]. Body temperature elevation peaked at about 60–70 min after LPS injection, with temperature increased by 1.06–1.78 °C from baseline and hyperthermia remained for 120 min and then declined with fluctuations. It is possible to use this sensor system to make a continuous measurement of the body temperature if one is aiming at developing new anti-inflammatory drugs and testing pyrogens in the future. As a preliminary study, the physical moving activity of rabbits was recorded with the sensor system. Rabbits were routinely fed in a single cage (44L × 50W × 40H cm in our facility). Figure 6 showed a representative result of the mobility of one rabbit within 36 h. We analyzed the correlation between body temperature and physical activity, but they were not positively correlated (Pearson correlation values are 0.14 and 0.15, n = 2). Due to the limited space within the rabbit cage, the current data on physical activity obtained seemed to be less informative, suggesting that it is necessary to compare these data with the real movement of rabbits monitored by a video camera in a large place in the future.

We noticed that when the telemetry sensor system is applied, the distance between the receiver-PC and antenna needs to be considered. The principle is “the nearer, the better” because the signals captured by the receiver is significantly reduced due to the presence of other radio wave interference in the same facility [22]. Our study showed that the signal loss was <5% in general if the receiver was placed in the animal room but the signal loss can be increased if the receiver was placed in different buildings, which is far from the animal room. Compared with other commercially available telemetry devices, the current sensor system may be limited in its use to mediate for large animals due to its size; therefore, it is necessary to develop more compact and smaller sensors to facilitate its use in small animals, such as mice [10,23].

It has been reported that the body temperature of rabbits is characterized by an inconspicuous circadian rhythm and always reaches the bottom in the morning and rises the peak during midnight [24,25], which was supported by our observations on three rabbits measured by the current sensor system. In addition to body temperature and physical activities, this sensor system is currently under modification and is expected to monitor body fluid composition, pH, and blood pressure in the future.

## 4. Conclusions

In conclusion, the simple telemetry sensor system was developed for monitoring the body temperature of rabbits. This sensor system will become a useful tool for measuring animal body temperature for testing new anti-inflammatory drugs.

## Figures and Tables

**Figure 1 animals-13-01677-f001:**
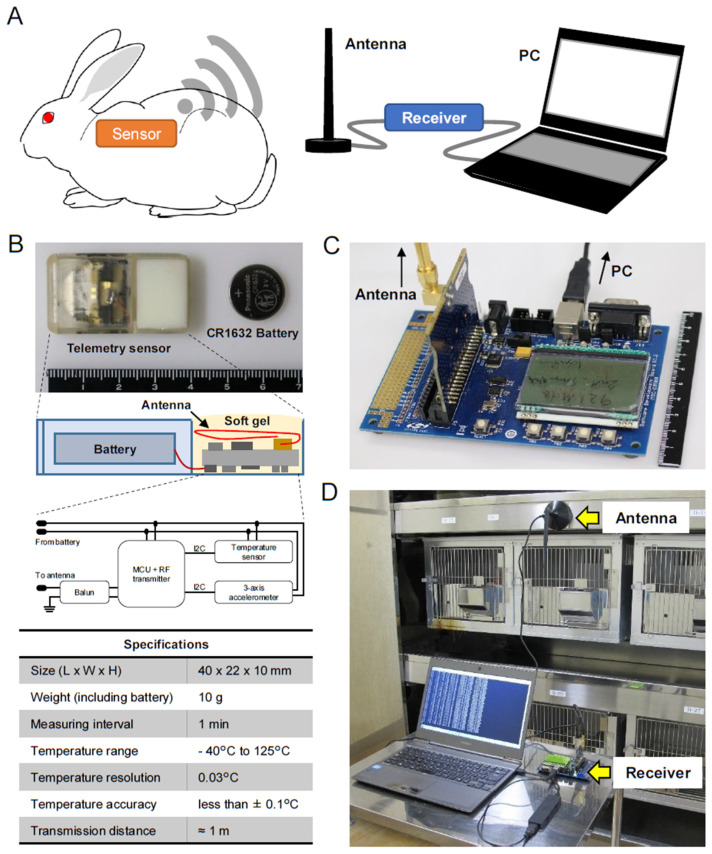
Telemetry sensor system. (**A**) Schematic illustration of the telemetry sensor system principle and basic parts. The wireless sensor implanted in the rabbits collects signals. The data was sent out through a radio transmitter and then captured by the receiver. The personal computer (PC) records and processes the transferred data. (**B**) The implantable telemetry sensor components and their specifications. The sensor is loaded with a battery and covered with biocompatible soft gel. (**C**) The picture of the receiver. (**D**) Outline of the entire setup of the telemetry system in the rabbit facility room. The antenna is installed on the rabbit cage rack and the receiver can simultaneously collect radio signals from several rabbits.

**Figure 2 animals-13-01677-f002:**
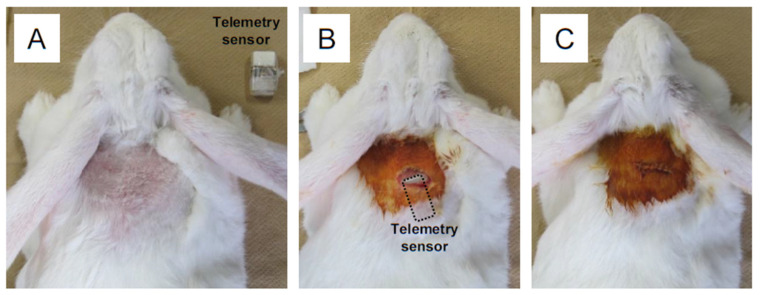
Surgical procedure for implanting a telemetry sensor in a rabbit. (**A**) the hair of the posterior cervical region of an anesthetized rabbit was shaved. (**B**) the skin was disinfected with povidone-iodine and a small incision (~3 cm) was made, and a telemetry sensor was implanted subcutaneously. The dashed line indicates the location of the implanted sensor. (**C**) The incised skin was sutured and disinfected.

**Figure 3 animals-13-01677-f003:**
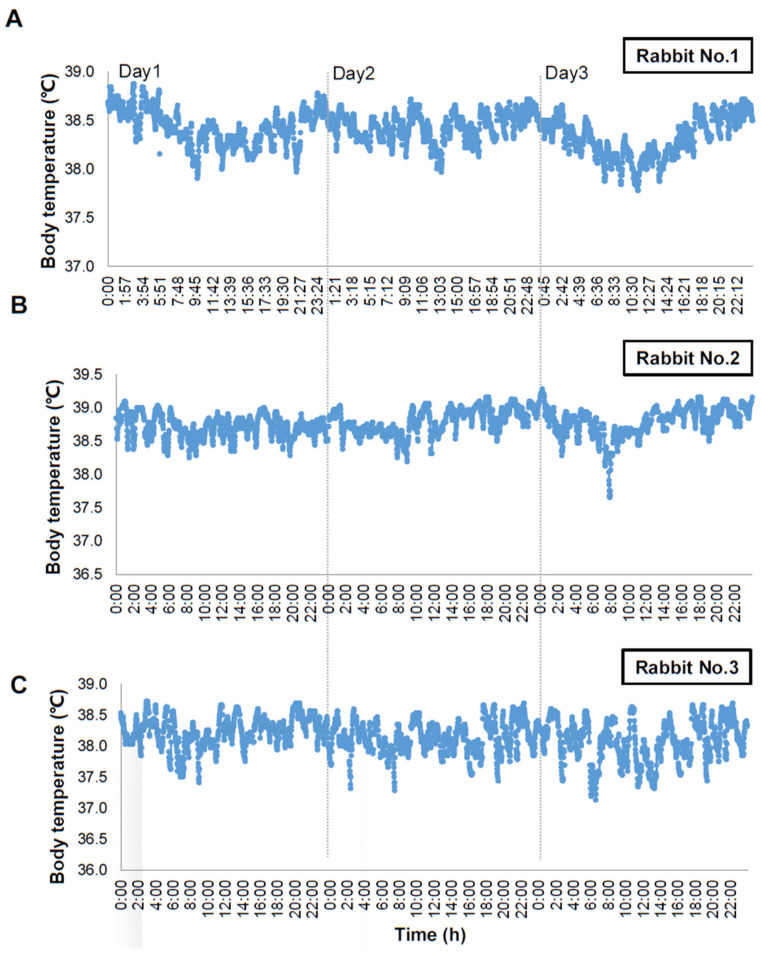
Representative data of real-time monitoring of the body temperature of three individual (designated as (**A**–**C**)) rabbits. The body temperature was measured with a subcutaneously implanted telemetry sensor at 1 min intervals for 3 consecutive days.

**Figure 4 animals-13-01677-f004:**
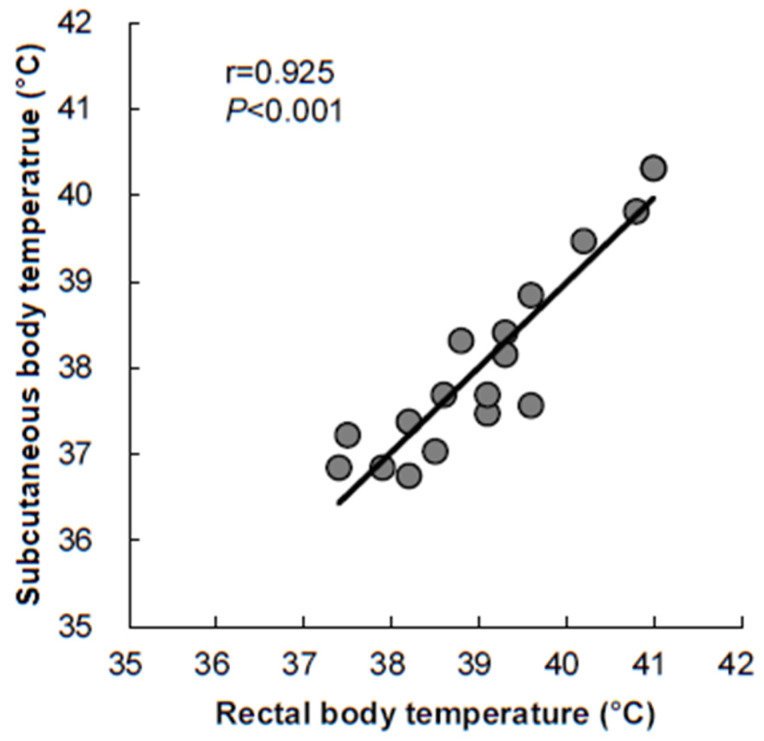
Correlation between subcutaneous and rectal temperature. Subcutaneous temperature was measured by the telemetry sensor whereas rectal temperature was measured by a digital thermometer under the normal condition. Data were collected at six time points from three rabbits.

**Figure 5 animals-13-01677-f005:**
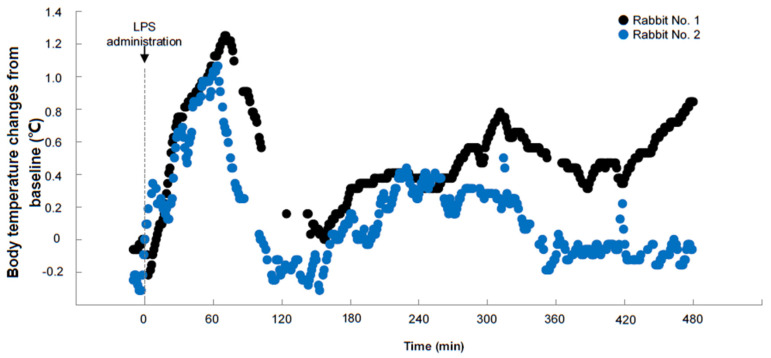
Body temperature changes after administration of a pyrogen. Lipopolysaccharides (LPS) was injected into two rabbits via an auricular vein (2 μg/kg BW). The body temperature was measured with the telemetry sensor system at 1 min interval. Changes of the body temperature were expressed as relative to the baseline.

**Figure 6 animals-13-01677-f006:**
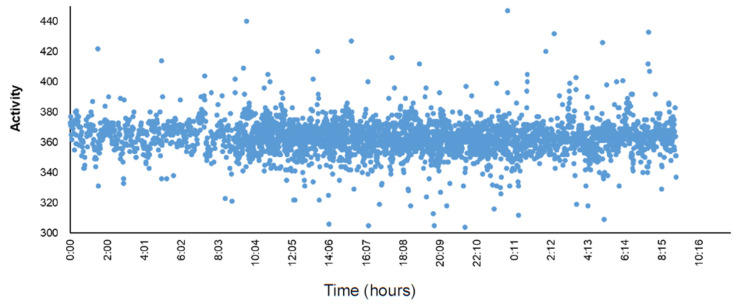
Representative data of physical moving activity of a rabbit measured by the telemetry sensor. The body temperature and activity were recorded simultaneously with the telemetry sensor system at 1 min interval for 30 consecutive hours. Three rabbits were measured, and physical activity is expressed as arbitrary values. X, Y, Z, activity is the data calculated using the following formula by an accelerometer as below. X: acceleration in X direction (in plane of the sensor node) Y: acceleration in Y direction (in plane of the sensor node, 90deg of X direction) Z: acceleration in Z direction (vertical to in plane. If the sensor node is face up, Z will be around 1024, and if face down Z will be −1024). Activity = $\frac{\sqrt{X^{2}+Y^{2}+Z^{2}}}{3}$, so the base line is around 330–340 (no movement).

## Data Availability

Data is available on request due to restrictions. The data presented in this study are available on request from the corresponding author. This telemetry sensor system can be supplied for interested researchers based on the collaborative agreement.

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
