# Peer review of "A Simple Telemetry Sensor System for Monitoring Body Temperature in Rabbits—A Brief Report"

_animals, 2023, doi:10.3390/ani13101677_

Round 1

Reviewer 1 Report

This paper starts by presenting state-of-the-art techniques for temperature monitoring in animals. The challenge is to develop a sensor system to measure subcutaneous temperatures in rabbits. The authors chose the radio frequency identification chip with the hybrid interface and neglectable power consumption, the technique is simple and not highly innovative but it is a good successful effort to monitor body temperature for mediate-sized animals for research applications. The paper is well written.
Just some aspects of improvement that the authors could consider:
A more rigorous and updated bibliographic review is recommended in the Introduction part that allows a better definition of the state of the art in the subject matter under study. The general and specific objectives of the study to be carried out must be indicated more clearly.
I suggest improving the discussion section with some advantages and disadvantages of an innovative model presented compared to other models analyzed with critical judgment.
The bibliography is scarce and should be reviewed and updated to improve the quality and interest of readers and researchers.
Specific comments:
Line 116: I suggest  “by a digital..”
Line 135: I suggest removing “as”
Line 135: “sensor”
Line 141: “ of pyrogen”
Line 143: “Changes in body temperature”
Line 144: “were expressed relative”
Line 179: “proven”
Line 185: “ camera”

Author Response

This paper starts by presewe nting state-of-the-art techniques for temperature monitoring in animals. The challenge is to develop a sensor system to measure subcutaneous temperatures in rabbits. The authors chose the radio frequency identification chip with the hybrid interface and neglectable power consumption, the technique is simple and not highly innovative but it is a good successful effort to monitor body temperature for mediate-sized animals for research applications. The paper is well written.

Just some aspects of improvement that the authors could consider:

A more rigorous and updated bibliographic review is recommended in the Introduction part that allows a better definition of the state of the art in the subject matter under study. The general and specific objectives of the study to be carried out must be indicated more clearly.

Thanks for your suggestion. We have modified the introduction and added some new bibliographic reviews in the revised manuscript. In addition, our objectives were emphasized.

I suggest improving the discussion section with some advantages and disadvantages of an innovative model presented compared to other models analyzed with critical judgment.

Thanks for your suggestion. We have made a comparison of our new sensor systems with other sensors regarding the advantages and disadvantages in the discussion (L176-187 and L195-199).

The bibliography is scarce and should be reviewed and updated to improve the quality and interest of readers and researchers.

Thanks for your kindly suggest and in the revised version, we add another 5 references and the list number as 1st, 2nd, 4th, 7th, 9th, 10th, 22nd, and 23th.

Specific comments:

Line 116: I suggest  “by a digital..”

Line 135: I suggest removing “as”

Line 135: “sensor”

Line 141: “ of pyrogen”

Line 143: “Changes in body temperature”

Line 144: “were expressed relative”

Line 179: “proven”

Thanks so much for your suggestion and we have corrected these words according to your suggestions.

Reviewer 2 Report

1. The method of validating the sensory system should be briefed in Abstract.

2. Similar miniature sensor systems are commercially available as addressed in Introduction, and the technique is neither innovative nor sophisticated. The authors shall provide persuasive information on the system developed in house. 

3. Please state why the rectal temperature can be used as a substitute to the hypothalamic temperature in the validation. 

4. All commercial or public-available goods should come with information of sourcing.

5. Please say something more on the built-in 3D accelerometer.

6. Please provide a framework of the software logic.

7. I see only one temperature trace in Figure 3.

8. Rabbit #3 is absent in Figure 5.

Author Response

  1. The method of validating the sensory system should be briefed in Abstract.

Thanks for your suggestion. We have added a sentence in the Abstract, L34-35. In addition, we have described validating method in detail in the materials and methods (L95-98).

  1. Similar miniature sensor systems are commercially available as addressed in Introduction, and the technique is neither innovative nor sophisticated. The authors shall provide persuasive information on the system developed in house.

Thanks for your critique. To clarify the novelty of our sensor system, we have described this point in the materials and methods (L86-90) with a new reference No.16.

  1. Please state why the rectal temperature can be used as a substitute to the hypothalamic temperature in the validation.

Thanks for your question. We have rephrased the sentence because the previous one is a little bit confusing. We wanted to say that the body temperature is controlled by temperature-sensitive neurons, called the temperature control center located in the hypothalamus, and body temperature can be monitored by measuring rectal temperature in animals such as cow, sleep, and canine, because it is a non-invasive measurement (46-48).

  1. All commercial or public-available goods should come with information of sourcing.

Thanks for your suggestion. We added all information in the revised manuscript.

  1. Please say something more on the built-in 3D accelerometer.

Thanks for your question. 3D accelerometer (ADXL362, Analog Devices, Inc. MA) can measure physical activity and body position in addition to temperature. We have added this information in the materials and methods (L93-95).

  1. Please provide a framework of the software logic.

Thanks for you question. The following is the software logic: activities of the rabbits were calculated by the data from accelerometer [Average of the square root of the sum of the squares of the acceleration values in the three directions (Activity =   )]. Temperature of the rabbits were measured directly by temperature sensor. To improve accuracy, 12 times measurement were done in a minute, and then the average data were sent to receiver wirelessly. 

  1. I see only one temperature trace in Figure 3.

Figure 3 contains temperature data of three individual rabbits (labelled as A, B, C).  Please see the final version of the figures.

  1. Rabbit #3 is absent in Figure 5.

Thanks for your question. After injection of LPS, the temperature of rabbit No.3 was remained at extremely high levels for 6 hours, which was apparently different from other rabbits in the current study and also our experience in the previous studies, therefore, we have omitted this rabbit data in Figure 5 since we did not have a clear explanation.

Reviewer 3 Report

Thank you for submitting your manuscript. Please answer the following queries that I have:

Page 3: “Body temperature changes were monitored by the sensor subcutaneously implanted in the posterior neck of rabbits (Fig.2).”: Why was this region of the body chosen?

Page 3: Figure 1 shows the racks of rabbit cages. Mention the dimensions of a cage.

Page 5: “A telemetry sensor disinfected..”. What disinfectant was used?

Page 5: “To compare the rectal temperature with temperature measured by the sensor on the neck, rectal temperature of the same conscious rabbits was measured using a digital thermometer”: How frequently was the rectal temperature taken?

Page 6: n = 3 rabbits. Not a sufficient number. Equal numbers of males and females should be included. Track any variation in body temperature between the sexes

Page 7: “Figure 3. Correlation between subcutaneous and rectal temperature. Subcutaneous temperature was measured by the telemetry sensor whereas rectal temperature was measured by a digital thermometer”. Figure 3 displays the graphs for 3 rabbits for body temperature (subcutaneous) against time.

Page 9: In Figure 6 what is activity measured in?

Page 10: “The peritoneal temperature is always higher than subcutaneous temperate and less affected by the environment, but measuring the peritoneal temperature is invasive.” Besides, the environment the author could mention other  reasons for variation in the subcutaneous temperature due to the body site chosen.

Author Response

Thank you for submitting your manuscript. Please answer the following queries that I have:

Page 3: “Body temperature changes were monitored by the sensor subcutaneously implanted in the posterior neck of rabbits (Fig.2).”: Why was this region of the body chosen?

Thanks for you question. The sensor was implanted in the posterior neck because it is easy to perform without influencing the activity of rabbits.

Page 3: Figure 1 shows the racks of rabbit cages. Mention the dimensions of a cage.

Thanks for you question. The size of cage is 44Lx50Wx40H cm as described in the results.

Page 5: “A telemetry sensor disinfected..”. What disinfectant was used?

Thanks for your question. The surface of silico was disinfected by 70% ethanol before implantation.

Page 5: “To compare the rectal temperature with temperature measured by the sensor on the neck, rectal temperature of the same conscious rabbits was measured using a digital thermometer”: How frequently was the rectal temperature taken?

Thanks for your question. The rectal temperature was measured every 6 hours in a day and averaged.

Page 6: n = 3 rabbits. Not a sufficient number. Equal numbers of males and females should be included. Track any variation in body temperature between the sexes

Thanks for your critique in this regard. In this study, only female rabbits were used according the 3R principle. The purpose of this work was to validate the accuracy of the telemetry sensor and usefulness of this system. However, we agree with you that it is necessary to compare other factors that may affect the temperature such as gender, age and other physiological conditions in future.

Page 7: “Figure 3. Correlation between subcutaneous and rectal temperature. Subcutaneous temperature was measured by the telemetry sensor whereas rectal temperature was measured by a digital thermometer”. Figure 3 displays the graphs for 3 rabbits for body temperature (subcutaneous) against time.

Thanks for your critique. Figure 4 (rather than Figure 3) should be correct and we apologize for this mistake.

Page 9: In Figure 6 what is activity measured in?

Thanks for your critique. In the current system, a 3D accelerometer was packaged in the sensor to measure moving activity.

Page 10: “The peritoneal temperature is always higher than subcutaneous temperate and less affected by the environment, but measuring the peritoneal temperature is invasive.” Besides, the environment the author could mention other reasons for variation in the subcutaneous temperature due to the body site chosen.

Thanks for your critique. We have mentioned this point in the discussion (L155-157).